# Effects of Yeast Culture on Laying Performance, Antioxidant Properties, Intestinal Morphology, and Intestinal Flora of Laying Hens

**DOI:** 10.3390/antiox13070779

**Published:** 2024-06-27

**Authors:** Quan Qiu, Zhichun Zhan, Ying Zhou, Wei Zhang, Lingfang Gu, Qijun Wang, Jing He, Yunxiang Liang, Wen Zhou, Yingjun Li

**Affiliations:** 1National Key Laboratory of Agricultural Microbiology, College of Life Science and Technology, Huazhong Agricultural University, Wuhan 430070, China; quanqiu@webmail.hzau.edu.cn (Q.Q.); hejingjj@mail.hzau.edu.cn (J.H.); fa-lyx@163.com (Y.L.); 2Wuhan Sunhy Biology Co., Ltd., Wuhan 430070, China; zzc@sunhy.cn (Z.Z.); tech@sunhy.cn (Y.Z.); zhangw8804@163.com (W.Z.); glf8254@163.com (L.G.); wqj@sunhy.cn (Q.W.); 3Green Chemical Reaction Engineering, Engineering and Technology Institute Groningen (ENTEG), University of Groningen, Nijenborgh 4, 9747 AG Groningen, The Netherlands

**Keywords:** yeast culture, laying hens, egg production, egg quality, intestinal mucosal structure, intestinal flora

## Abstract

Yeast culture (YC) plays a significant role in enhancing the performance and health of poultry breeding. This study investigated the impact of different YC supplementation concentrations (basal diet with 1.0 g/kg and 2.0 g/kg of YC, YC1.0, and YC2.0) on egg production performance, egg quality, antioxidant properties, intestinal mucosal structure, and intestinal flora of laying hens. Both YC1.0 and YC2.0 groups significantly enhanced the egg protein height, Haugh unit, and crude protein content of egg yolks compared to the control group (*p* < 0.05). The supplementation with YC2.0 notably increased the egg production rate, reduced feed-to-egg ratio, and decreased the broken egg rate compared to the control group (*p* < 0.05). Additionally, YC supplementation enhanced serum total antioxidant capacity (T-AOC) and glutathione peroxidase (GSH-PX) activity while reducing malondialdehyde (MDA) content (*p* < 0.05). Moreover, YC supplementation promoted duodenal villus height and villus ratio in the duodenum and jejunum (*p* < 0.05). Analysis of cecal microorganisms indicated a decrease in Simpson and Shannon indices with YC supplementation (*p* < 0.05). YC1.0 reduced the abundance of Proteobacteria, while YC2.0 increased the abundance of Bacteroidales (*p* < 0.05). Overall, supplementation with YC improved egg production, quality, antioxidant capacity, intestinal morphology, and cecal microbial composition in laying hens, with significant benefits observed at the 2.0 g/kg supplementation level.

## 1. Introduction

Antibiotics have historically played a crucial role in improving livestock and poultry production by reducing disease incidence and enhancing performance [1]; however, their overuse has led to many issues like antibiotic residues, the emergence of drug-resistant strains, and environmental contamination, posing significant risks to food safety and human health [2]. Consequently, many countries are phasing out or restricting antibiotic use in animal farming [3]. While this policy has enhanced food safety, it has also negatively impacted production efficiency, including decreased feed conversion rates and increased disease treatment costs [4]. Thus, finding effective antibiotic alternatives is imperative for sustaining the health and productivity of the livestock industry.

Yeast culture (YC) is a microecological preparation comprising various metabolites derived from yeast cells grown on a specific culture medium and fully fermented under controlled conditions [5]. YC plays a role in maintaining a healthy balance of gut microbiota and supporting the growth of beneficial bacteria in the gut to enhance digestive health [6]. It serves as a promising antibiotic alternative in its prebiotic effects and immune modulation. Comprising yeast cell contents, metabolites, fermentation substrates, and minor inactive cells, it is abundant in functional compounds such as amino acids, peptides, oligosaccharides, and vitamins, enhancing the immune response and making animals less susceptible to diseases that would otherwise require antibiotic treatment [6]. Widely utilized in ruminant, swine, and poultry farming, its supplementation enhances heat stress resistance in dairy cows [7], early lactation milk yield in dairy goats [8], reproductive performance in sows, the growth of weaned piglets [9], and immune function in broiler chickens [10].

Numerous studies have demonstrated the benefits of YC supplementation in reducing the need for antibiotics, like improved growth performance, enhanced disease resistance, and better production outcomes, indirectly contributing to improved parameters of egg quality [6,11]. Moreover, YC has many economic and environmental benefits in cost savings and sustainable farming [12]. Overall, YC offers a natural alternative to antibiotics by enhancing gut health, supporting the immune system, and improving overall animal performance, thus translating to consistent egg production and quality [13,14]. This approach not only addresses the growing concern of antibiotic resistance but also promotes better health and productivity in livestock, contributing to more sustainable and economically viable farming practices.

Current research predominantly focuses on utilizing YC in broiler diets, leaving a gap in understanding its impact on laying hens. This study investigates the effects of dietary YC supplementation on egg production, quality, antioxidant properties, intestinal mucosal structure, and the microbiota of laying hens. Findings aim to furnish valuable insights into the application of YC in laying hen diets, enhancing theoretical foundations for its widespread adoption.

## 2. Materials and Methods

### 2.1. Diets, Experimental Design, and Hens

Yeast culture (YC), sourced from Wuhan Sunhy Biological Co., Ltd. (Wuhan, China), is derived from the *Kluyveromyces marxianus* strain and various cereal substrates. Through controlled liquid–solid two-phase deep fermentation and low-temperature drying, the process generated bio-products abundant in small peptides, amino acids, organic acids, and mannans. The key nutritional indicators include the following: crude protein ≥16.0%; crude fiber ≤ 11.0%; crude ash ≤ 9.0%; moisture ≤ 11.0%; small peptides ≥ 30.0%; mannan ≥ 0.8%; lactic acid ≥ 3.0%.

A total of 300 healthy 40-week-old Hy-Line brown laying hens, with the same egg production rates and body weights, were selected using a single-factor completely randomized design. They were then randomly divided into three treatment groups, each with five replicates of twenty hens. The control group (control) was fed a corn–soybean meal basal diet formulated according to the National Research Council (NRC, 1994) nutritional requirements (refer to Table 1). Detailed information of premix is provided in Appendix A. The experimental groups, YC1.0 and YC2.0, were supplemented with 1.0 g/kg and 2.0 g/kg of YC, respectively, in the basal diet. The trial period lasted for 35 days. The final weight of each laying hen was approximately 1.8 kg, measured under standardized conditions at the end of the testing period.

The laying hen housing system comprises a three-tiered cage structure (length, width, and height are 100 cm, 50 cm, and 60 cm, respectively) with artificial and natural lighting, maintaining a consistent 15-h photoperiod (05:00 a.m.–20:00 p.m.) at an intensity of 10–15 Lux. Temperature (15–20 °C) and humidity (40–60%) were regulated by mechanical and natural ventilation systems. Feeding occurred twice daily (07:30 a.m. and 15:30 p.m.), with amounts adjusted to minimize leftover feed. Temperature and humidity levels were monitored at these times. Fecal management and routine immunizations were performed as standard protocols. Egg production rates were monitored to ensure statistical parity among groups. Continuous access to food and water was provided throughout the experiment. All experimental procedures were approved by the University’s Animal Care and Use Committee (SCXK2020-0084), and performed in accordance with internationally accepted guidelines and ethical principles.

### 2.2. Measurements of Laying Eggs

#### 2.2.1. The Laying Rate of Eggs

To assess the performance of egg production, total egg weight, egg count, feed consumption, remaining feed inventory, flock size, average egg weight, egg production rate (including broken eggs), average intake, and feed-to-egg ratio were daily measured.

Egg production rate (%) = Total number of eggs × 100/(number of days × number of laying hens);

Average egg weight (g) = Total egg weight/number of eggs;

Average daily feed intake (g) = Total feed intake × 100/(number of days × number of of laying hens);

Feed-to-egg ratio = Total feed consumption/total egg weight;

Broken egg rate (%) = Number of broken eggs × 100/total number of eggs.

#### 2.2.2. The Quality of Eggs

On the 35th day of the experiment, two eggs per group were randomly selected for comprehensive egg quality assessment. Measurements included egg weight, shape index, Haugh units (HU), eggshell weight, moisture, protein, and fat content of the egg white and yolk. The egg shape index was determined using vernier calipers with 0.01 mm precision, measuring the longitudinal and maximum transverse diameters. Eggshells were cleaned, dried overnight at 50–53 °C, and weighed. Protein content in the egg yolk and white was analyzed by Kjeldahl™ 8100 (FOSS, Hillerød, Denmark). The fat content was determined with a Soxhlet apparatus (Soxtherm SOX 406, Gerhardt, Königswinter, Germany). The parameters were measured as follows:

Egg shape index (%) = Transverse diameter × 100/longitudinal diameter;

Haugh units (HU) = 100 × LOG [H + 7.57 − 1.7 × w^0.37^] (H is the average height of concentrated albumen, w is the weight of the egg);

Eggshell gravity (%) = Eggshell dry weight × 100/egg weight;

Egg white/egg yolk moisture content (%) = (egg white/egg yolk wet weight − egg white/egg yolk dry weight) × 100/(egg white/egg yolk dry weight);

Egg white/egg yolk crude protein content (%) = c × v × 14 × 6.25 × 100/m (c is the concentration of the standard hydrochloric acid solution (mol/L); v is the volume of the standard hydrochloric acid solution used (mL); 14 is the molar mass of nitrogen (g/mol); 6.25 is the conversion factor from nitrogen to crude protein; mm is the mass of egg white or egg yolk (g));

Fat content (%) = (m_1_ − m_0_) × M_1_ × 100/(M_2_ × M_0_) (m_1_ represents the weight of the dried ether cup after fat extraction; m_0_ is the weight of the dried ether cup before extraction; M_1_ denotes the weight of the egg yolk; M_2_ indicates the weight of the egg yolk sample; and M_0_ signifies the initial weight of the egg. All the units are in grams);

Egg yolk gravity (%) = weight of egg yolk × 100/weight of whole egg.

### 2.3. Determination of Antioxidant Capacity

To analyze the serum antioxidant capacity, one hen from each replicate group was randomly selected at 07:00 am on day 36 of the experiment. Blood samples (4 mL) were drawn from the wing vein after 2-h water deprivation and centrifuged at 3000 rpm/min for 10 min to obtain serum. The supernatant was aliquoted into 1.5 mL centrifuge tubes and stored at −20 °C for analysis of serum antioxidant indicators. Serum total antioxidant capacity (T-AOC), malondialdehyde (MDA) content, superoxide dismutase (SOD) activity, and glutathione peroxidase (GSH-PX) activity were measured using kits from Nanjing Jiancheng Bioengineering Institute (Nanjing, China), following the instructions of the manufacturer.

### 2.4. Analysis of Intestinal Mucosal Structure

To investigate the intestinal mucosal structure, the duodenum, jejunum, and ileum were collected from one of the laying hens of each replicate group. A 1–2 cm section from the middle of each intestinal segment was placed in 4% paraformaldehyde solution. These intestinal segments were embedded in paraffin, sectioned (5 µm thick), and stained with hematoxylin–eosin (HE). Under a light microscope, 15 villi per section were randomly selected to measure villus height and crypt depth. Villus height was measured from the villus tip to the crypt entrance, and crypt depth from the crypt base to the submucosa. The villus height/crypt depth (V/C) ratio was calculated to assess intestinal morphology.

### 2.5. Analysis of Gut Microbiota

To investigate the influence of YC on the gut microbiota composition in laying hens, 16S rDNA high-throughput sequencing analysis was used. Cecal contents were collected from one of the laying hens per replicate group, placed in sterile 2 mL centrifuge tubes, rapidly frozen in liquid nitrogen, and stored at −80 °C for further analysis. Total DNA was extracted from the frozen cecal contents using the QIAamp Fast DNA Stool Kit (Qiagen, Hamburg, Germany). DNA quality was assessed by 0.8% agarose gel electrophoresis and quantified with a UV spectrophotometer. The V3 region of the 16S rRNA gene of the cecal microflora was sequenced using 454 high-throughput sequencing technology. Low-quality sequences were filtered out using Mothur software (Version 1.35.1), and sequences with >97% similarity were clustered into operational taxonomic units (OTUs). Bacterial community diversity was analyzed and compared between the control and treatment groups using the Silva and RDP databases. Sequencing and analysis were performed by Beijing Meiji Biotechnology company (Beijing, China).

### 2.6. Statistical Analysis

Experimental data were organized in Excel 2016, and a one-way analysis of variance (ANOVA) was performed using SPSS 16.0. Tukey’s HSD test was used for multiple comparisons. Results are presented as mean ± standard deviation, with significance set at *p* < 0.05.

## 3. Results

### 3.1. Effects of Yeast Culture (YC) Supplementation to Diet on Egg Production Performance of Laying Hens

Table 2 illustrates the effects of the yeast culture (YC) supplementation on laying hens. Adding 1.0 g/kg YC (YC1.0) significantly reduced the broken egg rate by 63.48% compared to the control (*p* < 0.05), without affecting egg production rate, egg weight, or daily feed intake (*p* > 0.05). Supplementation with 2.0 g/kg YC (YC2.0) significantly improved the egg production rate, feed-to-egg ratio, and broken egg rate (*p* < 0.05), with no significant changes in egg weight and daily feed intake (*p* > 0.05).

### 3.2. Effects of YC Supplementation to Diet on Egg Quality of Laying Hens

Table 3 demonstrates the effect of YC supplementation on egg quality parameters. Adding 1.0 g/kg of YC significantly improved albumen height by 9.73% and Haugh units by 4.69%, with no significant influence on the specific gravity of egg yolk, eggshell specific gravity, fat content, and egg yolk crude protein content compared to the control group. Supplementation with 2.0 g/kg of YC significantly increased albumen height by 10.32%, Haugh units by 4.98%, and egg yolk crude protein content by 4.27%, without affecting egg yolk specific gravity, eggshell specific gravity, fat content, and egg white protein content.

### 3.3. Effects of YC Supplementation to Diet on Serum Antioxidant Capacity of Laying Hens

Figure 1 shows the effects of YC supplementation on serum antioxidant parameters in laying hens, with detailed information provided in Appendix A. Supplementing the diet with 1.0 g/kg and 2.0 g/kg of YC significantly increased total antioxidant capacity (T-AOC) by 35.91% and 48.39%; glutathione peroxidase (GSH-PX) activity by 15.95% and 19.13%; and reduced malondialdehyde (MDA) content by 39.74% and 44.55% (*p* < 0.05). There was no significant effect on superoxide dismutase (SOD) activity compared to the control group (*p* > 0.05).

### 3.4. Effects of YC Supplementation to Diet on Intestinal Mucosal Structure of Laying Hens

Figure 2 illustrates the effects of YC supplementation on intestinal morphology in laying hens, with detailed data presented in Table 4. Supplementing with 1.0 g/kg and 2.0 g/kg kg/ton of YC significantly increased duodenal villus height by 17.91% and 22.48%; duodenal villus height to crypt depth ratio (V/C) by 21.93% and 30.08%; and jejunal V/C by 22.69% and 26.89% (*p* < 0.05). Meanwhile, there were no significant changes in jejunal villus height, ileal villus height, ileal V/C, and crypt depth in the duodenum, jejunum, and ileum (*p* > 0.05).

### 3.5. Effects of YC Supplementation to Diet on Cecal Microbiota Composition of Laying Hens

Figure 3 illustrates the influence of YC supplementation to diet on the cecal microbiota of laying hens. As shown in Figure 3A, YC supplementation significantly reduced the Simpson and Shannon indices (*p* < 0.05), indicating decreased microbial diversity. However, there was no significant effect on the Chao1 and ACE indices (*p* > 0.05). Figure 3B presents Principal Coordinate Axis Analysis (PCoA), which revealed no significant clustering differences between the experimental groups, indicating that YC supplementation did not significantly affect the β-diversity of the cecal microbial communities in laying hens.

Figure 3C,D, along with Appendix A, display the relative abundance of cecal microbiota at the phylum and genus levels. Figure 3C indicates that the predominant phyla are Bacteroidetes, Firmicutes, Actinobacteria, and Proteobacteria. Supplementation with 1.0 g/kg of YC significantly increased the abundance of Actinobacteria (*p* < 0.05) and reduced Proteobacteria (*p* < 0.05), while changes in Bacteroidetes and Firmicutes were not significant (*p* > 0.05). Supplementation with 2.0 g/kg of YC resulted in non-significant changes in these phyla (*p* > 0.05).

At the genus level, bacterial genera with relative abundance higher than 1% included *Bacteroidales*, *Bacteroides*, *Unclassified_S24-7*, *Ruminococcaceae*, *Unclassified_BS11*, *Clostridiales*, *Lactobacillus*, *Veillonellaceae*, *Prevotella*, and *Parabacteroides* (Figure 3D). Supplementation with 1.0 g/kg of YC increased the relative abundance of *Bacteroidales*, Unclassified_S24-7, *Lactobacillus*, and *Rikenellaceae* (*p* > 0.05), while reducing *Bacteroides*, *Clostridiales*, *Faecalibacterium* (*p* > 0.05), *Veillonellaceae*, *Oscillospira*, and *Ruminococcus* (*p* < 0.05). Supplementation with 2.0 g/kg of YC showed similar trends, with significant increases in *Bacteroidales* (*p* < 0.05) and reductions in *Veillonellaceae* and *Ruminococcus* (*p* < 0.05).

## 4. Discussion

### 4.1. Effects of YC Supplementation to Diet on Egg Production Performance of Laying Hens

Yeast culture (YC) is rich in peptides, amino acids, vitamins, oligosaccharides, organic acids, minerals, and other beneficial factors that enhance livestock and poultry growth by improving feed palatability and promoting nutrient digestion and absorption, thereby boosting production performance [6,15]. Studies have shown that dietary YC increases egg production rate and egg weight while reducing the feed-to-egg ratio in laying hens [16]. Zhang et al. mentioned that 0.3% YC supplementation for 4 weeks significantly enhanced egg production rate, reduced feed-to-egg ratio due to YC supplementation, and upregulated intestinal digestive enzyme activities and intestinal health-related gene expression [17]. Similarly, Liu et al. reported that 1% YC significantly increased the egg production rate and decreased the feed-to-egg ratio, thereby improving overall egg production performance [18].

In this study, supplementation with 2.0 g/kg of YC significantly increased egg production rate, reduced feed-to-egg ratio, and decreased broken egg rate, demonstrating that YC improved egg production performance under these conditions. Zhang et al. also demonstrated that YC supplementation significantly increased amylase and chymotrypsin activity in the duodenum, enhanced the expression of duodenal tight junction proteins, and improved intestinal structure [17]. Liu et al. reported that YC significantly improved the apparent digestibility of dietary crude protein, confirming that YC enhances feed nutrient digestibility, and, consequently, production performance in laying hens [18].

### 4.2. Effects of YC Supplementation to Diet on Egg Quality of Laying Hens

Egg quality encompasses the shape, size, cleanliness, eggshell strength, and content quality, including the consistency of egg white, yolk size and color, and fat content. Egg yolk concentration is a key nutritional aspect. Higher yolk-to-egg ratios indicate better quality, and Haugh units measure protein quality and freshness, serving as a primary international standard for egg quality assessment. In this experiment, YC supplementation significantly increased egg protein height, Haugh units, and the crude protein content of egg yolk. With higher YC levels, the specific gravity of egg yolk, eggshell, and fat content also increased, while the egg shape index as well as the moisture content of egg white and egg yolk remained unchanged. Studies by Zhong et al. [19], Özsoy et al. [16], and Liu et al. [6] similarly reported that YC enhances egg protein height and Haugh units. Liu et al. observed a dose-dependent increase in these parameters with an addition of YC. Zhong et al. [19] also noted significant increases in yolk and eggshell weight.

These improvements are attributed to enhanced digestion and nutrient deposition facilitated by YC, which is rich in small peptides, amino acids, vitamins, and other nutrients that promote metabolic processes in laying hens [20,21]. However, some studies, such as Yalçin et al. [22] and Zhang et al. [17], found no significant effects on egg protein height, Haugh units, or eggshell quality, likely due to variations in yeast species, supplementation levels, hen age, and management conditions.

### 4.3. Effects of YC Supplementation to Diet on Serum Antioxidant Capacity of Laying Hens

Glutathione peroxidase (GSH-Px) and superoxide dismutase (SOD) are essential antioxidant enzymes that remove free radicals, protect cells from oxidative damage, and aid in cell repair [23]. Malondialdehyde (MDA) is a marker of lipid peroxidation, indicating oxidative stress levels [24]. The total antioxidant capacity (T-AOC) reflects the overall antioxidant defense against oxidative damage. Normally, free radical production and elimination are balanced but stress increases free radical production, leading to oxidative damage and higher MDA levels. To counter this, animals produce endogenous antioxidants, including enzymes like GSH-Px and SOD. Measuring these enzymes and MDA levels in serum assesses the antioxidant capacity of livestock and poultry.

Research indicates that YC enhances antioxidant enzyme activity and reduces MDA levels, improving overall antioxidant capacity [15]. Liu et al. found that dietary YC significantly increased serum T-AOC activity in laying hens but did not affect GSH-Px activity and MDA levels [6]. In contrast, Liu et al. reported that YC significantly increased serum GSH-Px and SOD activities while reducing MDA levels [18].

In this experiment, dietary YC significantly increased serum T-AOC and GSH-Px activities and reduced MDA levels in laying hens. SOD activity also showed an upward trend with higher YC levels. These findings align with previous studies, indicating that YC enhances antioxidant capacity and reduces oxidative damage. This effect is attributed to the rich oligosaccharide content in YC, particularly mannan oligosaccharides, which possess strong antioxidant and free-radical scavenging abilities [25,26]. Additionally, mannan oligosaccharides and glucans improve antioxidant properties by enhancing intestinal mucosa structure and promoting the absorption of nutrients like zinc, selenium, and copper [27].

### 4.4. Effects of YC Supplementation to Diet on Intestinal Mucosal Structure of Laying Hens

The small intestine is the primary site for nutrient digestion and absorption in animals. Key indicators of intestinal function include villus height, crypt depth, and the villus height-to-crypt depth ratio. Higher villi indicate more absorptive epithelial cells, enhancing nutrient absorption; shallower crypts suggest more mature cells with stronger secretion capabilities. A higher villus height-to-crypt depth ratio indicates a better intestinal mucosal structure and stronger digestion and absorption capacity [28]. Research demonstrates that YC improves intestinal structure, protects mucosa, and promotes development. Gao et al. found that 2.5 g/kg of YC significantly increased duodenal villus height and the villus-to-crypt ratio in broiler chickens [29]. He et al. reported that 2.0% of YC significantly enhanced jejunal and ileal villus height and the villus-to-crypt ratio in pigs [26]. Liu et al. showed similar effects in weaned piglets [30], and Zhang et al. noted increased jejunal villus height and villus-to-crypt ratio in geese with 2% YC supplementation [31].

In this experiment, dietary YC significantly increased duodenal villus height and the villus-to-crypt ratio in the duodenum and jejunum, consistent with previous studies. Mannan oligosaccharides in YC inhibited pathogenic bacterial colonization, protecting intestinal mucosa, and maintaining its normal function [32]. They also promoted beneficial bacteria growth, which ferment oligosaccharides to produce organic acids like acetic, propionic, and butyric acids. These acids support probiotic bacteria growth and provide energy for intestinal epithelial cells, promoting mucosal development [11]. Additionally, small peptides of YC have enhanced intestinal epithelial cell proliferation and inhibit apoptosis, aiding intestinal development according to the research from Wang et al. [33]. YC also supplies nucleotides necessary for rapid growth, promoting intestinal cell division, differentiation, and maturation [34]. Moreover, Sindaye et al. found that lysozyme could enhance intestinal morphology in laying hens [35].

### 4.5. Effects of YC Supplementation to Diet on Cecal Microbiota Composition of Laying Hens

The balance of intestinal microflora is crucial for animal health. Under normal conditions, the intestinal flora maintains a dynamic equilibrium, with bacterial abundances remaining stable. External stimuli can disrupt this balance, leading to an imbalance in intestinal flora. In the cecum of laying hens, the dominant bacterial phyla were Firmicutes, Bacteroidetes, Proteobacteria, and Actinobacteria, as previously reported [36,37]. During peak egg production, healthy hens showed 47.5–62.0% of Bacteroidetes and 30.8–60.4% of Firmicutes, based on reporting by others [38,39,40].

The previous studies indicate that dietary YC increases Bacteroidetes and decreases Firmicutes and Proteobacteria in the cecum of laying hens [6,18]. Bacteroidetes include probiotic bacteria involved in protein and polysaccharide degradation, enhancing nutrient metabolism and egg production as reported by Panasevich et al. [41] and Xu et al. [42]. Proteobacteria, which includes many pathogens like *Escherichia coli* and *Salmonella*, decrease in abundance with YC, promoting intestinal health according to the reports from Bi et al. [43]. Meanwhile, Tian et al. mentioned that the regulation of cecal microbiota might result in the production performance and egg quality [44].

The previous studies indicate that *Bacteroides* (11.1–32.5%), *Riken* (10.1–15.0%), *Lachnospira* (5.0–8.0%), and *Lactobacillus* (2.0–10.0%) dominate the cecum of laying hens at the genus level [38,39,40]. YC oligosaccharides promote probiotic growth and inhibit harmful bacteria, modulating intestinal flora composition. Specifically, YC increases the abundance of *Lactobacillus*, although not always significantly, suggesting a positive impact on intestinal health and hen growth [6,19,45].

## 5. Conclusions

This study revealed that adding yeast culture (YC) to the diet can improve the composition of intestinal flora and intestinal mucosal structure of laying hens and improve antioxidant capacity, thereby improving the production performance and egg quality of laying hens. Adding 2.0 g/kg of YC had an impact on production performance and egg quality, antioxidant capacity, intestinal mucosal structure, and bacterial flora structure better than at the addition level of 1.0 g/kg.

## Figures and Tables

**Figure 1 antioxidants-13-00779-f001:**
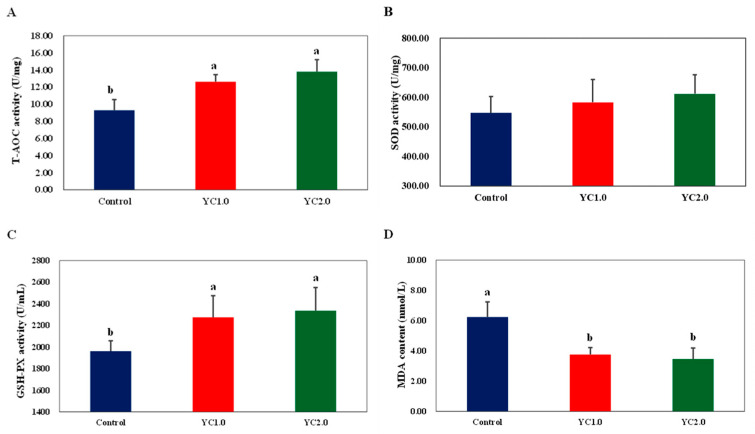
The effect of YC supplementation on serum antioxidant capacity of laying hens. (**A**) is total antioxidant capacity (TOC) activity, (**B**) is superoxide dismutase (SOD) activity, (**C**) is glutathione peroxidase (GSH-PX) activity, and (**D**) is malondialdehyde (MDA) content. ^a,b^ Labels in a row, the differences are significant (*p* < 0.05); labels containing the same lowercase letters or no letters, the differences are not significant (*p* > 0.05).

**Figure 2 antioxidants-13-00779-f002:**
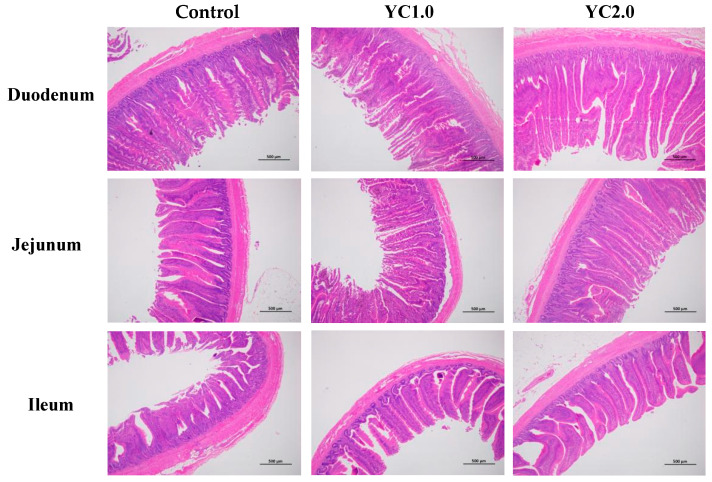
The effect of YC supplementation on intestinal mucosal structure of laying hens.

**Figure 3 antioxidants-13-00779-f003:**
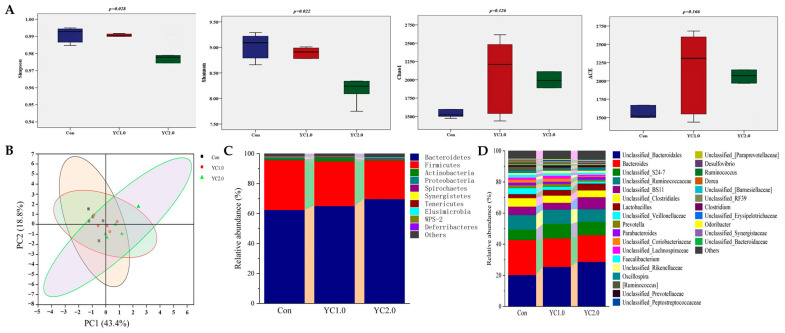
The effect of YC supplementation to diet on cecal microbiota of laying hens. (**A**) is the box plots of α diversity index. (**B**) is Principal Coordinate Axis Analysis (PCoA) two-dimensional sorting diagram. (**C**) is the relative abundance of cecal microflora at phylum level. (**D**) is the relative abundance of cecal microflora at genus level.

**Table 1 antioxidants-13-00779-t001:** The composition and nutrient levels of basal diets (air-dried basis).

Items	Content (%)
Ingredients	
Corn	60.7
Soybean	23.5
Soybean oil	1
CaCO_3_	9
CaHPO_4_	0.8
Premix	5
Total	100
Nutrient content	
Metabolizable energy (MJ/kg)	10.93
Crude protein	16.09
Lysine	0.78
Methionine and Cystine	0.52
Calcium	3.55
Available Phosphorus	0.52

**Table 2 antioxidants-13-00779-t002:** Effect of yeast culture (YC) supplementation on laying performance of laying hens.

Items	Control	YC1.0	YC2.0	SEM	*p* Value
Egg laying rate (%)	90.25 ± 1.36 ^b^	91.78 ± 1.25 ^ab^	92.62 ± 1.33 ^a^	0.408	0.042
Average daily feed intake (g)	127.49 ± 3.87	124.56 ± 4.46	123.61 ± 3.13	1.023	0.290
Average egg weight (g)	61.54 ± 0.64	62.43 ± 0.30	62.05 ± 0.96	0.190	0.162
Ratio of feed to egg	2.29 ± 0.03 ^a^	2.19 ± 0.06 ^a^	2.15 ± 0.09 ^b^	0.022	0.019
Broken egg rate (%)	1.15 ± 0.23 ^a^	0.42 ± 0.23 ^b^	0.26 ± 0.18 ^b^	0.116	<0.001

Data are means of 5 replicates of 20 samples each replicate. SEM, standard error of mean. ^a,b^ Labels in a row, the differences are significant (*p* < 0.05); labels containing the same lowercase letters or no letters, the differences are not significant (*p* > 0.05).

**Table 3 antioxidants-13-00779-t003:** Effect of yeast culture (YC) supplementation on egg quality of laying hens ^1^.

Items	Control	YC1.0	YC2.0	SEM	*p* Value
Shape index	1.32 ± 0.06	1.32 ± 0.04	1.32 ± 0.06	0.003	0.842
Yolk ratio (%)	27.84 ± 1.15	28.11 ± 1.58	28.97 ± 1.26	0.346	0.412
Eggshell ratio (%)	10.31 ± 0.21	10.48 ± 0.23	10.58 ± 0.83	0.126	0.693
Albumen height (mm)	6.78 ± 0.30 ^b^	7.44 ± 0.34 ^a^	7.48 ± 0.30 ^a^	0.114	0.007
Haugh unit	83.17 ± 1.96 ^b^	87.07 ± 1.51 ^a^	87.31 ± 1.98 ^a^	0.669	0.006
Egg white moisture (%)	87.46 ± 0.40	87.82 ± 0.53	87.37 ± 0.34	0.115	0.250
Egg yolk moisture (%)	45.81 ± 0.35	45.37 ± 0.35	44.99 ± 1.04	0.183	0.186
Fat content (%)	8.08 ± 0.37	8.34 ± 0.49	8.49 ± 0.52	0.121	0.402
Protein content in egg white (%)	9.54 ± 0.32	9.54 ± 0.37	9.99 ± 0.24	0.094	0.067
Protein content in egg yolk (%)	16.85 ± 0.18 ^b^	17.28 ± 0.55 ^ab^	17.57 ± 0.34 ^a^	0.121	0.039

^1^ Data are means of 5 replicates of 20 samples each replicate. SEM, standard error of mean. ^a,b^ Labels in a row, the differences are significant (*p* < 0.05); labels containing the same lowercase letters or no letters, the differences are not significant (*p* > 0.05).

**Table 4 antioxidants-13-00779-t004:** Effect of YC supplementation on intestinal mucosal structure of laying hens.

Items		Control	YC1.0	YC2.0	SEM	*p* Value
Duodenum	Villus height (µm)	1040.39 ± 93.32 ^b^	1226.71 ± 140.77 ^a^	1274.25 ± 108.21 ^a^	38.651	0.018
	Crypt depth (µm)	139.77 ± 9.07	134.41 ± 10.59	131.04 ± 8.34	2.439	0.363
	Villus/Crypt	7.48 ± 0.99 ^b^	9.12 ± 0.73 ^a^	9.73 ± 0.70 ^a^	0.321	0.003
Jejunum	Villus height (µm)	833.49 ± 120.03	932.63 ± 121.38	982.61 ± 90.40	31.386	0.141
	Crypt depth (µm)	140.95 ± 13.69	129.24 ± 21.76	130.20 ± 8.72	4.006	0.447
	Villus/Crypt	5.95 ± 0.94 ^b^	7.30 ± 1.08 ^a^	7.55 ± 0.57 ^a^	0.285	0.031
Ileum	Villus height (µm)	573.08 ± 106.65	659.34 ± 107.01	648.33 ± 102.28	27.185	0.399
	Crypt depth (µm)	109.67 ± 19.07	103.45 ± 8.61	101.48 ± 9.94	3.330	0.612
	Villus/Crypt	5.33 ± 1.20	6.39 ± 0.99	6.40 ± 0.91	0.282	0.217

Data are means of 5 replicates of 20 samples each replicate. SEM, standard error of mean. ^a,b^ Labels in a row, the differences are significant (*p* < 0.05); labels containing the same lowercase letters or no letters, the differences are not significant (*p* > 0.05).

## Data Availability

The original contributions presented in the study are included in the article/Supplementary Material, further inquiries can be directed to the corresponding author.

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
