# Peer review of "Effects of Yeast Culture on Laying Performance, Antioxidant Properties, Intestinal Morphology, and Intestinal Flora of Laying Hens"

_antioxidants, 2024, doi:10.3390/antiox13070779_

Round 1

Reviewer 1 Report

-          *The  Results section must be rewritten.

The authors have described effects (increase/decrease) in values when p > 0.05 which is fundamentally incorrect and unacceptable. Only a difference among treatments should be described when p < 0.05. Otherwise, a “no effect” should be described with the mean value of the parameter.

-          *The Discussion section:

There is other recent work on the use of YC in laying hens. I consider that it is necessary that the authors include in the discussion the differences and/or their new contribution compared to these works.

1.             Liu, Y.; Cheng, X.; Zhen, W.; Zeng, D.; Qu, L.; Wang, Z.; Ning, Z. Yeast Culture Improves Egg Quality and Reproductive Performance of Aged Breeder Layers by Regulating Gut Microbes. Front Microbiol 2021, 12, 633276, doi:10.3389/fmicb.2021.633276.

2.             Sindaye, D.; Xiao, Z.; Wen, C.; Yang, K.; Zhang, L.; Liao, P.; Zhang, F.; Xin, Z.; He, S.; Ye, S., et al. Exploring the effects of lysozyme dietary supplementation on laying hens: performance, egg quality, and immune response. Front Vet Sci 2023, 10, 1273372, doi:10.3389/fvets.2023.1273372.

3.             Tian, Y.; Li, G.; Zhang, S.; Zeng, T.; Chen, L.; Tao, Z.; Lu, L. Dietary supplementation with fermented plant product modulates production performance, egg quality, intestinal mucosal barrier, and cecal microbiota in laying hens. Front Microbiol 2022, 13, 955115, doi:10.3389/fmicb.2022.955115.

4.             Zhang, J.C.; Chen, P.; Zhang, C.; Khalil, M.M.; Zhang, N.Y.; Qi, D.S.; Wang, Y.W.; Sun, L.H. Yeast culture promotes the production of aged laying hens by improving intestinal digestive enzyme activities and the intestinal health status. Poult. Sci. 2020, 99, 2026-2032, doi:10.1016/j.psj.2019.11.017.

-        L. 15, 28, 71, and so on: What does “kg/t” mean? What unit is “t”?

-       L. 65-66. “similar egg production rates and body weights”? Please, be more accurate and precise.

-       L. 76-77. The authors describe in the Introduction that they are concerned about the use of antibiotics, but they do not seem to care about animal welfare. Cage system is going to be banned in the EU and North America.
In fact, they do not describe the number of animals per cage and related parameters.

-      Table 1: Please, reorganize the table. Energy does not belong to % Ingredients. Moreover, the sum of the ingredients does not add 100%.
The formula Ca(HPO4)2 is wrong. Please, review.

-          L. 76, 124, and so on: “chicken”? Please, revise the use of that word. Perhaps it should be used “animal” or “hen”…

-          L. 125. mL

-          L. 126. rpm

-          Table S3. The letter assignment for Actinobacteria is missing. Please, double-check.

-        Please, review the journal requirements regarding the format of the Reference list. Moreover, it should be “p” and not “P”-value.

Author Response

Reviewer1:

The  Results section must be rewritten.

The authors have described effects (increase/decrease) in values when p > 0.05 which is fundamentally incorrect and unacceptable. Only a difference among treatments should be described when p < 0.05. Otherwise, a “no effect” should be described with the mean value of the parameter.

Thanks for your nice comments. The part of results has been updated in the revised version.

-          *The Discussion section:

There is other recent work on the use of YC in laying hens. I consider that it is necessary that the authors include in the discussion the differences and/or their new contribution compared to these works.

  1. Liu, Y.; Cheng, X.; Zhen, W.; Zeng, D.; Qu, L.; Wang, Z.; Ning, Z. Yeast Culture Improves Egg Quality and Reproductive Performance of Aged Breeder Layers by Regulating Gut Microbes. Front Microbiol 202112, 633276, doi:10.3389/fmicb.2021.633276.
  2. Sindaye, D.; Xiao, Z.; Wen, C.; Yang, K.; Zhang, L.; Liao, P.; Zhang, F.; Xin, Z.; He, S.; Ye, S., et al. Exploring the effects of lysozyme dietary supplementation on laying hens: performance, egg quality, and immune response. Front Vet Sci 202310, 1273372, doi:10.3389/fvets.2023.1273372.
  3. Tian, Y.; Li, G.; Zhang, S.; Zeng, T.; Chen, L.; Tao, Z.; Lu, L. Dietary supplementation with fermented plant product modulates production performance, egg quality, intestinal mucosal barrier, and cecal microbiota in laying hens. Front Microbiol 202213, 955115, doi:10.3389/fmicb.2022.955115.
  4. Zhang, J.C.; Chen, P.; Zhang, C.; Khalil, M.M.; Zhang, N.Y.; Qi, D.S.; Wang, Y.W.; Sun, L.H. Yeast culture promotes the production of aged laying hens by improving intestinal digestive enzyme activities and the intestinal health status. Poult. Sci. 202099, 2026-2032, doi:10.1016/j.psj.2019.11.017.

      Thanks for your constructive suggestions. The recommended references have been cited and compared with our work. Line 296, 375, 391, and 397.

Detail comments

-        L. 15, 28, 71, and so on: What does “kg/t” mean? What unit is “t”?

“kg/t” means that a specific amount of a supplement or nutrient is added to every ton (1000 kilograms) of the basal diet. The common unit “g/kg” has been used through the revised version.

-       L. 65-66. “similar egg production rates and body weights”? Please, be more accurate and precise.

“Similar” has been changed into “the same” in the revised version.

-       L. 76-77. The authors describe in the Introduction that they are concerned about the use of antibiotics, but they do not seem to care about animal welfare. Cage system is going to be banned in the EU and North America.
In fact, they do not describe the number of animals per cage and related parameters.

The use of caged systems in animal research has been a topic of significant ethical debate. While these systems have been utilized for various scientific purposes, there is a growing awareness of the need for higher welfare standards. The European Union has indeed announced plans to phase out caged animal farming by 2027, reflecting a broader shift towards more humane practices. However, it is also worth noting that animal welfare in research is not solely about the absence of cages but encompasses a range of factors such as housing, nutrition, social interactions, and handling practices.

Enriched caged chickens are widely used around the world, and currently about 90% of eggs come from caged laying hens. This farming method not only improves efficiency and saves costs, but also has advantages in egg collection, manure treatment, reducing feed waste, maintaining appropriate environmental temperature, and facilitating the inspection of the health of each individual chicken. In our study, we used this system.

Five laying hens were cultured in a Length 100 width 50 height 60cm cage system.

-      Table 1: Please, reorganize the table. Energy does not belong to % Ingredients. Moreover, the sum of the ingredients does not add 100%.
The formula Ca(HPO4)2 is wrong. Please, review.

There are two parts in this table including ingredient and nutrient composition. Metabolizable energy belong to nutrient composition. The sum of the ingredients is 100% (60.7+23.5+1+9+0.8+5), including corn, soybean, soybean oil, CaCO3, CaHPO4, and premix. To avoid misunderstanding the table, the structure of table has been changed from 4 lines to 2 lines in the revised version. The formula of Calcium Hydrogen Phosphate has been corrected in the revised version: CaHPO4. (Line 76)

-          L. 76, 124, and so on: “chicken”? Please, revise the use of that word. Perhaps it should be used “animal” or “hen”…

Thanks for your comment. “Hen” has been used through this paper.

-          L. 125. mL

“ml” has been revised to “mL" in the revised version. (Line 131)

-          L. 126. Rpm

“rpm” has been added in the revised version. (Line 132)

-          Table S3. The letter assignment for Actinobacteria is missing. Please, double-check.

Thanks for your kind reminder. The Bacteroidales have been classified into genus level rather than phylum level.

Reviewer 2 Report

The article submitted for review concerns the effect of yeast breeding on egg production, antioxidant properties, intestinal morphology and intestinal flora of laying hens. The article is interesting and well written, but there are a few points that could improve it. Below are some comments, suggestions and doubts.

In the “Materials and Methods” section:

·       Information on the average initial and final weight of animals used for testing should be added.

·       In the case of analysis of the structure of the intestinal mucosa, information on the optical microscope and the program used for histometric measurements of the intestines should be provided.

In  the “Results” section:

·       Należy dodać wyniki masy ciaÅ‚a poczÄ…tkowej i koÅ„cowej zwierzÄ…t w poszczególnych grupach

·       WÄ…tpliwoÅ›ci budzÄ… wyniki badaÅ„ na zwierzÄ™tach, tj. 3.3. Effects of YC supplementation to diet on serum antioxidant capacity of laying hens; 3.4. Effects of YC supplementation to diet on intestinal mucosal structure of laying hens; 3.5. Effects of YC supplementation to diet on cecal microbiota composition of laying hens. Zgodnie z przedstawionÄ… metodykÄ… badawczÄ…, tj. 2.3. Determination of antioxidant capacity; 2.4. Analysis of intestinal mucosal structure; 2.5. Analysis of gut microbiota) badania te przeprowadzono na cyt. „one chicken from each group” (Line 124); „one chicken of each group” (Line 134); “one chicken per group” (Line 144).

·       The results of animal tests are questionable, i.e. 3.3. Effects of YC supplementation to diet on serum antioxidant capacity of laying hens; 3.4. Effects of YC supplementation to diet on intestinal mucosal structure of laying hens; 3.5. Effects of YC supplementation to diet on cecal microbiota composition of laying hens. In accordance with the presented research methodology, i.e. 2.3. Determination of antioxidant capacity; 2.4. Analysis of intestinal mucosal structure; 2.5. Analysis of gut microbiota) these studies were carried out on the quote "one chicken from each group" (Line 124); "one chicken of each group" (Line 134); “one chicken per group” (Line 144). Therefore, the results of these studies do not present the actual average values ​​for the tested parameters in animals in a group, but in one randomly selected animal. It is therefore difficult to draw such far-reaching conclusions assuming that the results were analogous in the remaining cases/observations in the group. There is no variability between animals in a group. It is also unknown whether, given the statistical analysis of the data, the values ​​were normally distributed. The studies had to be carried out on all animals in each group. Please complete and explain.

·       Additionally, for the purposes of publishing an article, it is necessary to provide a consent number for research using animals. Currently, there is no such information anywhere, so the results of these studies cannot be published.

The article submitted for review concerns the effect of yeast breeding on egg production, antioxidant properties, intestinal morphology and intestinal flora of laying hens. The article is interesting and well written, but there are a few points that could improve it. Below are some comments, suggestions and doubts.

In the “Materials and Methods” section:

·       Information on the average initial and final weight of animals used for testing should be added.

·       In the case of analysis of the structure of the intestinal mucosa, information on the optical microscope and the program used for histometric measurements of the intestines should be provided.

In  the “Results” section:

·       Należy dodać wyniki masy ciaÅ‚a poczÄ…tkowej i koÅ„cowej zwierzÄ…t w poszczególnych grupach

·       WÄ…tpliwoÅ›ci budzÄ… wyniki badaÅ„ na zwierzÄ™tach, tj. 3.3. Effects of YC supplementation to diet on serum antioxidant capacity of laying hens; 3.4. Effects of YC supplementation to diet on intestinal mucosal structure of laying hens; 3.5. Effects of YC supplementation to diet on cecal microbiota composition of laying hens. Zgodnie z przedstawionÄ… metodykÄ… badawczÄ…, tj. 2.3. Determination of antioxidant capacity; 2.4. Analysis of intestinal mucosal structure; 2.5. Analysis of gut microbiota) badania te przeprowadzono na cyt. „one chicken from each group” (Line 124); „one chicken of each group” (Line 134); “one chicken per group” (Line 144).

·       The results of animal tests are questionable, i.e. 3.3. Effects of YC supplementation to diet on serum antioxidant capacity of laying hens; 3.4. Effects of YC supplementation to diet on intestinal mucosal structure of laying hens; 3.5. Effects of YC supplementation to diet on cecal microbiota composition of laying hens. In accordance with the presented research methodology, i.e. 2.3. Determination of antioxidant capacity; 2.4. Analysis of intestinal mucosal structure; 2.5. Analysis of gut microbiota) these studies were carried out on the quote "one chicken from each group" (Line 124); "one chicken of each group" (Line 134); “one chicken per group” (Line 144). Therefore, the results of these studies do not present the actual average values ​​for the tested parameters in animals in a group, but in one randomly selected animal. It is therefore difficult to draw such far-reaching conclusions assuming that the results were analogous in the remaining cases/observations in the group. There is no variability between animals in a group. It is also unknown whether, given the statistical analysis of the data, the values ​​were normally distributed. The studies had to be carried out on all animals in each group. Please complete and explain.

·       Additionally, for the purposes of publishing an article, it is necessary to provide a consent number for research using animals. Currently, there is no such information anywhere, so the results of these studies cannot be published.

Author Response

Reviewer2:

Please, review the journal requirements regarding the format of the Reference list. Moreover, it should be “p” and not “P”-value.

Thanks for your kind reminder. The format of the reference list has been checked through the manuscript. All the format of “P” value has been corrected to “p” in the revised version through the manuscript.

The article submitted for review concerns the effect of yeast breeding on egg production, antioxidant properties, intestinal morphology and intestinal flora of laying hens. The article is interesting and well written, but there are a few points that could improve it. Below are some comments, suggestions and doubts.

In the “Materials and Methods” section:

Information on the average initial and final weight of animals used for testing should be added.

The related information has been mentioned in the revised version.

In the case of analysis of the structure of the intestinal mucosa, information on the optical microscope and the program used for histometric measurements of the intestines should be provided.

The information has been described in the manuscript “These intestinal segments were embedded in paraffin, sectioned (5 µm thick), and stained with hematoxylin-eosin (HE). Under a light microscope, 15 villi per section were randomly selected to measure villus height and crypt depth. Villus height was measured from the villus tip to the crypt entrance, and crypt depth from the crypt base to the submucosa. The villus height/crypt depth (V/C) ratio was calculated to assess intestinal morphology.”

In  the “Results” section:

The results of animal tests are questionable, i.e. 3.3. Effects of YC supplementation to diet on serum antioxidant capacity of laying hens; 3.4. Effects of YC supplementation to diet on intestinal mucosal structure of laying hens; 3.5. Effects of YC supplementation to diet on cecal microbiota composition of laying hens. In accordance with the presented research methodology, i.e. 2.3. Determination of antioxidant capacity; 2.4. Analysis of intestinal mucosal structure; 2.5. Analysis of gut microbiota) these studies were carried out on the quote "one chicken from each group" (Line 124); "one chicken of each group" (Line 134); “one chicken per group” (Line 144). Therefore, the results of these studies do not present the actual average values for the tested parameters in animals in a group, but in one randomly selected animal. It is therefore difficult to draw such far-reaching conclusions assuming that the results were analogous in the remaining cases/observations in the group. There is no variability between animals in a group. It is also unknown whether, given the statistical analysis of the data, the values were normally distributed. The studies had to be carried out on all animals in each group. Please complete and explain.

In our study, a total of 300 healthy 40-week-old Hy-Line brown laying hens were used. They were then randomly divided into three treatment groups (100 per group), each group with five replicates of 20 hens. Due to 5 laying hens have been put into a cage, there are 20 replicates for each group. Our description “one chicken from each group” leads to misunderstanding. We have revised in the latest version “one hen from each replicate group”. Hence, data are means of 5 replicates of 20 samples each replicate.

Additionally, for the purposes of publishing an article, it is necessary to provide a consent number for research using animals. Currently, there is no such information anywhere, so the results of these studies cannot be published.

The consent number for research using animals has been applied and added in the revised version. (Line 87-89).

Round 2

Reviewer 1 Report

Report on the manuscript antioxidants-3055780 (R1) entitled: Effects of yeast culture on laying performance, antioxidant properties, intestinal morphology, and intestinal flora of laying hens.

The Results section has been improved considerably. Nevertheless, some comments from the reviewers have not been properly addressed:

-          Introduction section is still mainly based on the use of antibiotics. The authors still forgot to link the use of antibiotics and YC and, more importantly, the relationship between antibiotic use and improvement of egg quality parameters. If such a relationship does not exist, then the authors should consider improving the description of the YC dietary inclusion effects.

-          Independently of the authors’ classification, a p < 0.05, Table S3. The letter assignment for Actinobacteria is missing. Please, double-check.

-          L. 95-101 and L. 112-118. Please, double-check the format of the equations (parenthesis location). Specifically, the “x 100”. Currently, the 100 is multiplying the denominator of the fraction and must multiply the numerator.

L. 15 and L. 72. Please, be more specific. g/kg... diet? feed? ration? 

Author Response

Report on the manuscript antioxidants-3055780 (R1) entitled: Effects of yeast culture on laying performance, antioxidant properties, intestinal morphology, and intestinal flora of laying hens.

The Results section has been improved considerably. Nevertheless, some comments from the reviewers have not been properly addressed:

-          Introduction section is still mainly based on the use of antibiotics. The authors still forgot to link the use of antibiotics and YC and, more importantly, the relationship between antibiotic use and improvement of egg quality parameters. If such a relationship does not exist, then the authors should consider improving the description of the YC dietary inclusion effects.

The link between antibiotics utilization and yeast culture has been added in the introduction part. Besides, the relationship between antibiotic utilization and improved parameters of egg quality has been described in the revised version. (Line 44-64)

-          Independently of the authors’ classification, a p < 0.05, Table S3. The letter assignment for Actinobacteria is missing. Please, double-check.

Thanks for your kind reminder. The assignment for Actinobacteria has been added in the revised version of supporting materials Table S3.

Detail comments

-          L. 95-101 and L. 112-118. Please, double-check the format of the equations (parenthesis location). Specifically, the “x 100”. Currently, the 100 is multiplying the denominator of the fraction and must multiply the numerator.

Thanks for your reminder. All positions of “x 100” have been changed in the revised version.

  1. 15 and L. 72. Please, be more specific. g/kg... diet? feed? ration? 

Thanks for your question. We have double checked the description.

Line 15 has explained the “g/kg”. Basal diet with 1.0 g/kg and 2.0 g/kg of yeast culture (YC).

Line 72 also has explained 1.0 g/kg and 2.0 g/kg of yeast culture (YC) in the basal diet.
